# Advancements in Vaccine Adjuvants: The Journey from Alum to Nano Formulations

**DOI:** 10.3390/vaccines11111704

**Published:** 2023-11-09

**Authors:** Sivakumar S. Moni, Siddig Ibrahim Abdelwahab, Aamena Jabeen, Mohamed Eltaib Elmobark, Duaa Aqaili, Gassem Ghoal, Bassem Oraibi, Abdulla Mohammed Farasani, Ahmed Ali Jerah, Mahdi Mohammed A. Alnajai, Abdul Majeed Hamad Mohammad Alowayni

**Affiliations:** 1Department of Pharmaceutics, College of Pharmacy, Jazan University, Jazan 45142, Saudi Arabia; aamenajabeen@gmail.com (A.J.);; 2Medical Research Centre, Jazan University, Jazan 45142, Saudi Arabiaboraibi@jazanu.edu.sa (B.O.); 3Physiology Department, Faculty of Medicine, Jazan University, Jazan 45142, Saudi Arabia; 4Department of Pediatrics, Faculty of Medicine, Jazan University, Jazan 45142, Saudi Arabia; 5College of Applied Medical Sciences, Jazan University, Jazan 45142, Saudi Arabia; 6General Directorate of Health Services and University Hospital, Jazan University, Jazan 45142, Saudi Arabia; mnajai@jazanu.edu.sa

**Keywords:** vaccine adjuvants, mineral salts, virus-like particles, liposomes, nanoparticles, nano vesicles

## Abstract

Vaccination is a groundbreaking approach in preventing and controlling infectious diseases. However, the effectiveness of vaccines can be greatly enhanced by the inclusion of adjuvants, which are substances that potentiate and modulate the immune response. This review is based on extensive searches in reputable databases such as Web of Science, PubMed, EMBASE, Scopus, and Google Scholar. The goal of this review is to provide a thorough analysis of the advances in the field of adjuvant research, to trace the evolution, and to understand the effects of the various adjuvants. Historically, alum was the pioneer in the field of adjuvants because it was the first to be approved for use in humans. It served as the foundation for subsequent research and innovation in the field. As science progressed, research shifted to identifying and exploiting the potential of newer adjuvants. One important area of interest is nano formulations. These advanced adjuvants have special properties that can be tailored to enhance the immune response to vaccines. The transition from traditional alum-based adjuvants to nano formulations is indicative of the dynamism and potential of vaccine research. Innovations in adjuvant research, particularly the development of nano formulations, are a promising step toward improving vaccine efficacy and safety. These advances have the potential to redefine the boundaries of vaccination and potentially expand the range of diseases that can be addressed with this approach. There is an optimistic view of the future in which improved vaccine formulations will contribute significantly to improving global health outcomes.

## 1. Introduction

The contribution of the English physician Edward Jenner (1749–1823) to eradicating smallpox is well known around the world. His findings have served society and resulted in the development of a new concept called a vaccine. The term vaccine is derived from “vacca”, meaning cow. Edward Jenner had observed that dairy maids were protected from smallpox. It had been recorded that Jenner heard from a dairy maid, “I shall never have smallpox for I have had cow pox. I shall never have an ugly pockmarked face”. This prompted Jenner to develop a method that saved many lives, and he termed his method vaccination. Edward Jenner said, “I hope that someday the practice of producing cow pox in human beings will spread over the world—when that day comes, there will be no more smallpox”. He was true to his word, and his work is widely recognized as the foundation of immunology [1]. Vaccines represent one of the most remarkable breakthroughs in preventing infectious diseases, primarily due to their ability to trigger a robust immune response against pathogens. Over time, the concept of vaccines has evolved from solely serving as prophylactic measures to also encompassing therapeutic vaccines. The critical role of vaccines in protecting populations from a range of diseases is well documented [2,3]. Conversely, the costs associated with vaccines and vaccination programs are a critical factor in the health policies of nations worldwide. The World Health Organization (WHO) has a strategy that emphasizes the development of vaccines that are both cost-effective and highly effective in the fight against various diseases [4]. However, the decision to incorporate a new vaccine into a country’s Expanded Programme on Immunization (EPI) or National Immunization Schedule is influenced by various circumstances, as discussed by Sivakumar et al. (2011) [4]. These considerations include factors such as the prevalence and impact of the disease, the patterns and mechanisms of disease transmission, and critical properties of vaccines such as their safety, effectiveness, cost, benefits, and availability.

An effective vaccine operates by stimulating the innate immune system and inducing a robust immune response. As a result, adjuvants have been added to vaccine formulations to enhance the body’s immune response. The practice of using adjuvants dates to the 1920s, when scientists discovered that adding substances such as aluminum salts to vaccines could enhance the effectiveness of the immune response [5]. Pashine et al. (2005) propose that potent adjuvants can bolster vaccine effectiveness through several mechanisms [6]. These include accelerating the development of a strong immune response, prolonging immune activity by establishing a depot at the injection site, and prompting plasma cells to generate antibodies with heightened binding affinity and improved pathogen-neutralizing capabilities, potentially activating cytotoxic T lymphocytes (CTLs) in the process [5]. This indicates that adjuvant-containing vaccines might be efficacious even for individuals with limited prior pathogen exposure. Furthermore, such adjuvants can magnify response rates in people with compromised immune systems, reduce the requisite antigen count—potentially enabling single-shot vaccinations—and thus potentially curtail the overall expenses associated with vaccination initiatives, notably benefiting less developed and developing nations.

The immune system functions with a dual nature that has evolved. It can trigger responses within minutes when it encounters molecular patterns associated with microbes, and it also develops a long-lasting immune response. The innate immune response acts as the first protective barrier, supported by receptors recognizing pathogen-associated molecular patterns (PAMPs). These receptors, called pattern recognition receptors (PRRs), are found in various cell types. They include immune cells such as neutrophils, macrophages, dendritic cells, natural killer cells, B cells, and non-immune cells such as epithelial and endothelial cells. Thus, activation of the innate immune system plays a critical role in the development of the acquired immune response [7,8,9,10]. Therefore, the incorporation of immune enhancers that elicit a robust and durable immune response is of paramount importance.

Historically, there have been concerns about the safety of vaccine adjuvants. Regulatory agencies such as the U.S. Food and Drug Administration (FDA) and the European Medicines Agency (EMA) enforce strict standards for the approval of new adjuvants. Ongoing studies aim to achieve a balance between efficacy and safety, with the goal of enhancing the immune response while reducing adverse effects [11]. New technologies, including nanoparticles and innovative chemical compounds, are being investigated for their suitability as next-generation adjuvants [12]. This article reviews the historical development and advances in adjuvant systems, from traditional forms such as alum to modern options such as nanoparticles. It provides a chronological account and highlights how the field of adjuvants has evolved over time in line with scientific discoveries and technological innovations. Nanoparticles offer a variety of benefits, including improved delivery methods and the ability to elicit a more targeted immune response. These new technologies represent a broader trend in science to develop adjuvants that are not only effective, but also have improved safety characteristics. This article provides a comprehensive overview of the historical development and current innovations in adjuvant systems, from the reliable use of alum to the exciting possibilities presented by nanoparticle-based techniques.

## 2. Adjuvants and Their Importance

The primary objective of vaccination is to safeguard individuals from diseases that can be prevented by inducing a robust and durable immune response. To achieve this goal, an adjuvant—derived from the Latin term “adjuvare”, meaning to support or enhance—is used as a carrier of the antigen. Thus, an adjuvant is a component, consisting of chemical substances or biomolecules, that enhances the specific immune response, thereby increasing the efficacy of vaccines [4]. In 1920, Gaston Ramon, a veterinarian from France, made a significant discovery while conducting research at the Pasteur Institute. Ramon successfully demonstrated that the addition of various ingredients such as breadcrumbs or flour to the inactivated diphtheria toxin led to local inflammation at the injection site and at the same time increased the formation of antibodies in response to the vaccine. This concept evolved into the foundational principle for enhancing the immune response of vaccines by incorporating specific substances [5]. In 1926, Glenny and colleagues demonstrated the significance of alum adjuvants in the context of diphtheria toxoids. Their findings not only highlighted the potential benefits of enhancing the immune response to various vaccines, but also paved the way for the development of vaccines with augmented immunogenicity. Building upon their discoveries, vaccines were created by inducing precipitation within alum. Thus, adjuvants are included in vaccine formulations to enhance the immunological response to the vaccine [4]. Effective adjuvants have the potential to improve vaccine efficacy via several mechanisms: eliciting specific immune responses, prolonging their duration, increasing the avidity and affinity of antibodies produced, stimulating cytotoxic T lymphocyte (CTL) responses, increasing response rates in individuals with lower responsiveness, and benefiting immunocompromised patients.

Numerous researchers have extensively highlighted the significance and advantages of incorporating adjuvant systems into vaccine formulations [13,14,15,16,17,18,19,20,21]. Over the past nine decades, many adjuvants have been developed to enhance the effective delivery of antigens in their original form. Nonetheless, not all these adjuvants have been approved for human use due to concerns surrounding their toxicity or inefficacy. Consequently, a diverse array of adjuvants with distinct mechanisms of action have emerged, encompassing mineral salts, emulsion-based adjuvants, immune-stimulating complexes (ISCOMs), biomolecules sourced from bacteria, natural substances, and innovative delivery systems involving polymers or liposomes [4]. The realm of vaccine development has advanced significantly through biotechnological research, notably with the advent of recombinant DNA technology. The first recombinant vaccine, targeting Hepatitis B, was generated utilizing recombinant DNA technology, and expressed within *Saccharomyces cerevisiae* [22]. Although recombinant peptide vaccines and subunit vaccines are generally safer compared to traditional vaccines, they often exhibit suboptimal immunogenicity [4]. This discrepancy has spurred the quest for robust and safe adjuvant systems, particularly with the goal of devising single-shot vaccines, a milestone advancement in human healthcare. In this context, polymeric nano/microparticles have demonstrated their merit as superior adjuvants, capable of inducing both humoral and cell-mediated immune responses.

In 2007, Aguilar and Rodriguez addressed the benefits of adjuvants, highlighting several important points. First, they underscored the critical role of adjuvants in enhancing the immune response by facilitating the presentation of antigens to the immune system in their unmodified form [23]. This approach is crucial for maximizing the efficacy of immune activation. Second, they emphasized the urgent need to develop lean immunization protocols that can be performed in a single step. This stems from the need to optimize the Expanded Programme on Immunization (EPI) with the dual goals of reducing immunization costs and reducing logistical challenges associated with maintaining a cold chain for vaccine storage and distribution. In addition, Aguilar and Rodriguez underscored the importance of adjuvants in stimulating CD8+ immune responses [23]. This aspect is particularly important in improving the immune status of immunocompromised individuals, thus contributing to the overall immunization strategy. In essence, the findings of Aguilar and Rodriguez highlight the multifaceted role of adjuvants in enhancing immune responses, facilitating vaccination protocols, and boosting immunity in different populations, thus making a notable contribution to the field of immunization [23]. Over the past nine decades, aluminum and calcium salts have been the only adjuvants approved for use in humans [4,24]. Historically important adjuvants have been superseded by MF 59 and ASO4, both of which have received approval for human use. These newer adjuvants are considered key components in modern vaccine technology. Despite the development of numerous adjuvants, none of them have received approval for human use, primarily due to concerns about increased toxicity [4]. Therefore, ensuring vaccine safety is a priority to protect the public from potentially serious toxicities resulting from routine vaccination. Given this primary concern, a final vaccine adjuvant should have several key properties: it must be nontoxic, biodegradable, cost-effective, nonimmunogenic, and capable of delivering antigens in their original form. These comprehensive criteria underscore the importance of both safety and efficacy in the ongoing search for optimal vaccine adjuvants. Over the past 90 years, a range of adjuvants with different modes of action have been formulated. These adjuvants can be divided into several groups: mineral salts, oil emulsions, immunostimulatory complexes (ISCOMs), immunopotentiators such as purified bacterial derivatives, and various components of bacterial cells, carbohydrates, peptides, and cytokines. In addition, particulate adjuvants such as virus-like particles, liposomes, and polymeric micro- and nanoparticles have been developed and classified.

### 2.1. Traditional Adjuvants

#### 2.1.1. Mineral Salts

Traditionally, mineral salts such as aluminum and calcium compounds have been used as vaccine adjuvants. Apart from these compounds, iron and zirconium salts have been screened to adsorb antigens [25,26].


*a.* 
*Aluminum Compounds*



Adjuvants in vaccines designed to stimulate immunity include aluminum hydroxide, aluminum phosphate, and potassium aluminum sulfate (commonly known as alum), as indicated in Table 1. Alum, referred to chemically as potassium aluminum sulfate, was initially employed for the purification of protein antigens [4]. Subsequently, researchers uncovered that the efficacy of alum-precipitated vaccines is greatly influenced by the presence of anions such as bicarbonates, sulfates, or phosphates [27]. Based on the manufacturing difficulties and therapeutic incompatibilities of alum-precipitated vaccines, Maschmann et al. (1931) reported the ability of aluminum hydroxide gel to adsorb protein antigens from an aqueous solution in a well-defined standardized method (Figure 1); thus, the preparation was termed an “alum-adsorbed vaccine” [28].

On the journey of vaccine adjuvant preparation and utilization, in 1946, Ericsson developed a method that demonstrated diphtheria toxoid co-precipitation using aluminum phosphate gel [29]. However, in 1947, Holt demonstrated aluminum phosphate as an adsorbent for diphtheria toxoid [30]. Adsorbing to an aluminum carrier is a challenge; various researchers have demonstrated that aluminum hydroxide is a better adjuvant than aluminum phosphate. This may be due to the better adjuvant capacity of the antigen at neutral pH with aluminum hydroxide when compared to aluminum phosphate [31,32].

**Table 1 vaccines-11-01704-t001:** Types of alum adjuvants for suitable vaccines.

Aluminum Adjuvant	Characteristics	Examples of Vaccines	References
Aluminum hydroxide	White, gel-like substance; insoluble in water	Hepatitis A, Hepatitis B, Human Papillomavirus	[31,33,34]
Aluminum phosphate	White, crystalline powder	Anthrax, Diphtheria-Tetanus-Pertussis, Haemophilus influenzae type b	[31,33,34]
Amorphous aluminum hydroxy phosphate sulfate (AAHS)	Nanoparticulate adjuvant, composed of very small particles	Human Papillomavirus, Hepatitis B	[31,33,34]

Lindblad and Sparck (1987) demonstrated the adsorption capacity of aluminum hydroxide. They reported that aluminum hydroxide adsorbed serum albumin 10 to 20 times more than aluminum phosphate [35]. Moreover, increasing the concentration of aluminum may lead to cytotoxicity to antigen-presenting cells. The US code of Federal Regulations governs the concentration of aluminum in a single dose of human vaccine. The permissible level of aluminum is 1.25 mg per dose of biological products for human use as per World Health Organization guidelines [36]. During the 1930s, the adjuvant properties of alum adjuvants for human vaccines were established, especially in toxoid vaccines for both diphtheria and tetanus. Thereafter, the use of aluminum by vaccine manufacturers became common across the world.

The major advantage of aluminum-adsorbed vaccines is the development of a high antibody titer even after primary immunization [37,38]. This is mainly due to the formation of a short-term depot at the site of injection, slowly releasing antigens and easily phagocytosed, which increases immune mechanisms. However, many adverse reactions have been reported, such as erythema, granulomas when the vaccine is administered in subcutaneous or intra-dermal routes rather than via intramuscular injection [39,40], and contact hypersensitivity [36,41]. Many researchers have reported that aluminum hydroxide is attracted towards recruiting eosinophil, which leads to an IgE-mediated allergic reaction at the site of injection and neurotoxicity [31,42,43,44,45,46]. In contrast, Gupta et al. (1995) stated that aluminum adjuvants have been used for many years for the hypo-sensitization of allergic patients without adverse effects [31]. In addition, aluminum consumption from vaccines is much less than that received from the diet or medications such as antacids [47]. However, under normal physiological conditions, aluminum is excreted through the kidneys, but under renal impairment conditions, aluminum accumulated in the body can lead to severe toxicities such as neurological symptoms, Alzheimer’s disease, amyotrophic lateral sclerosis, and dialysis-associated dementia [46]. Although aluminum adjuvants have been approved by the US FDA for human use, they still have some limitations, such as lacking the ability to elicit a cytotoxic T lymphocyte (CTL) response, especially for viral antigens. Earlier reports showed that aluminum adjuvants failed to provoke a satisfactory immune response against influenza and typhoid vaccines [48].

Furthermore, aluminum adjuvants have shown limited applicability for small peptides, subunits, and recombinant proteins. Despite this limitation, aluminum adjuvants have been screened for few DNA vaccine formulations. Reports suggest that aluminum hydroxide inhibits the effect of a vaccine, whereas aluminum phosphate augments the vaccine immune response [49,50]. Another limitation is that aluminum adjuvants cannot be frozen or lyophilized easily [51,52].


*b.* 
*Calcium phosphate*



Calcium phosphate is used as a vaccine adjuvant like aluminum compounds (Table 2).

Calcium phosphate is another mineral salt that has been licensed as an adjuvant in Europe and has been used in European DTP vaccines for many years [56]. Calcium phosphate adjuvant was developed by the Institute Pasteur, Paris, as tricalcium phosphate (Ca_3_(PO_4_)_2_ form [57]. It has the potential advantages of being a normal constituent of the body, well tolerated, readily reabsorbed, and not increasing IgE production [58,59,60,61]. Relyveld et al. (1985 and 1986) demonstrated calcium phosphate as a vaccine adjuvant. Calcium phosphate has been used successfully as an adjuvant for diphtheria, tetanus, polio, BCG, yellow fever, measles, and hepatitis B vaccines [56,62,63]. The adsorption capacity of calcium phosphate is highly influenced by factors such as the electrostatic attraction between the vaccine antigen and calcium phosphate [64,65]. pH plays a significant role that dictates the adsorption properties; hence, it should comply with the normal physiological value for human vaccination [31]. Studies suggest that there is no relation between antigenicity and vaccine adsorption to the adjuvant. Calcium phosphate gel has exhibited greater antigenicity than aluminum hydroxide in mice. The antigenicity is not associated with adsorption capacity, since aluminum hydroxide has shown better adsorption capacity [66]. However, the antigenicity varies with the physical state of the adjuvant. The gel form of calcium phosphate has higher antigenicity properties [45,66]. Adverse effects are usually rare and mild with calcium-phosphate-adsorbed vaccines. Calcium phosphate has also been used with HIV-l gp160 antigen in rabbits [67]. Jiang (2004) reported that neurological reactions to pertussis vaccines adsorbed to calcium phosphate were rarely observed [64]. Calcium phosphate gel shows high hemolytic activity if the concentration is higher than 1.3 mg/mL, and hence, the adverse effect of calcium phosphate gel is dose dependent. Therefore, the WHO and European Pharmacopoeia reported that the concentration of calcium phosphate should not exceed 1.3 mg/mL per vaccine dose [68,69]. In 2016, Yousef Amini et al. developed calcium phosphate nanoparticles and reported good immunization against tuberculosis because tubercle antigen can be easily adsorbed on calcium phosphate nanoparticles and lead to a strong cellular immune response [70]. Although calcium phosphate has been replaced by alum-type adjuvants, it persists as a vaccine adjuvant approved by the World Health Organization (WHO) for human vaccination [68].

#### 2.1.2. Oil Emulsion Adjuvants

Emulsions, mixtures of two or more liquid phases, have a long history of use as vaccine adjuvants, serving to enhance the immune response to antigens [71,72]. These are colloidal systems that improve immunogenicity through a “depot effect”, which prolongs the interaction between antigens and immune cells, as well as triggering an inflammatory response that activates the immune system. The earliest use of emulsions for this purpose dates to 1916, when Le Moignic and Pinoy showed that a water-in-Vaseline oil emulsion containing dead Salmonella typhimurium cells could elicit a heightened immune response [73].


*a.* 
*Freund’s Adjuvants*



Freund’s adjuvants, named after Jules T. Freund, who formulated the first one in 1937, are a set of substances specifically designed to amplify the immunogenic effects of antigens. Freund and his colleagues devised an innovative water-in-oil (*w*/*o*) emulsion using paraffin oil as the oil component [74,75]. This type of emulsion has been shown to significantly enhance the immune response to the encapsulated antigen. In 1956, Freund introduced two variations of the water-in-oil emulsion: Freund’s complete adjuvant (FCA), which contained dead mycobacteria, and incomplete Freund’s adjuvant (IFA), which did not contain any mycobacteria. Both forms have been instrumental in boosting the efficacy of vaccines [76].



*Complete Freund’s Adjuvant (CFA)*



CFA is a *w*/*o* type of emulsion prepared using paraffin oil as an external phase and mixed with desiccated mycobacteria in water as an internal phase. CFA reported as an adjuvant showed a strong immune response by enhancing both humoral and cell-mediated immunity [75]. Due to the presence of dead mycobacteria in CFA, it attracts macrophages and other phagocytic cells to the injection site as an innate immune phase that is followed by an adaptive immune phase. However, CFA has been reported to induce toxic effects that lead to lesions, granulomas, and inflammation at the injection site [77].



*Incomplete Freund’s adjuvant (IFA)*



IFA is a *w*/*o* type of emulsion prepared using paraffin oil without dead mycobacteria. IFA forms a short-term depot at the site of injection, releasing the antigen in a sustained manner and attracting immune cells. However, IFA has shown poor immunomodulatory effects because it lacks dead mycobacteria [76]. Studies have suggested that IFA used in human influenza vaccine exhibits better antibody titers when compared to influenza vaccine without IFA. Louis et al. (2005) repeated the use of IFA as a potential adjuvant for malarial vaccine [78]. The World Health Organization (WHO) reported that IFA has shown severe adverse effects, such as sterile abscesses at the site of injection [76].


*b.* 
*Montanide*



Montanide is a family of adjuvants used in the development of vaccines and other immunological applications. These adjuvants are designed to enhance the immune response to an antigen and make vaccines more effective. They can be formulated as oil-in-water or water-in-oil emulsions prepared using mineral oil. The adjuvant has been screened for both prophylactic and therapeutic human vaccines. Montanide ISA 720 is a special adjuvant of the Montanide family that is commonly used in vaccine formulations. It is often formulated as a water-in-oil (*w*/*o*) emulsion and is used primarily in animal vaccines, although it has been studied for human applications. This adjuvant is intended to enhance the immune response to the antigen of the vaccine, thereby improving the overall efficacy of the vaccine. The water-in-oil emulsion of Montanide ISA 720 provides a “depot effect”, a slow-release system that allows the antigen to be presented to the immune system for a longer period. This prolonged interaction enhances the body’s immune response. In addition, Montanide ISA 720 is thought to trigger an inflammatory response, further activating the immune system to respond more strongly to the antigen [79]. Montanide ISA 720 has been widely used in veterinary applications; its use in human vaccines is generally more restricted and subject to more stringent regulatory evaluations. Montanide ISA 51 is a specific adjuvant in the Montanide series that is commonly used in vaccine formulations to enhance the immune response. This adjuvant is usually in the form of an oil-in-water (*o*/*w*) emulsion, but other Montanide formulations may be water-in-oil (*w*/*o*) emulsions [80]. An experimental study suggested that both Montanide ISA 720 and Montanide ISA 51 VG exhibited good induction of Th_2_ (CD4+) and CTL (CD8+) cells to enhance both humoral and cellular immune responses [80,81].


*c.* 
*Adjuvant 65-4*



Adjuvant 65-4 is a specific formulation of a mineral-oil-based adjuvant that was developed by Merck & Co, USA in the 1960s. It was initially designed to improve the efficacy of influenza vaccines by enhancing the immune response. While Adjuvant 65-4 was an important early effort in the field of vaccine development, it is less commonly used today in human vaccines due to safety concerns [82,83].

*d.* 
*MF 59*


MF59 is an oil-in-water emulsion containing 4.3% squalene, a triterpene steroid derived from shark liver oil. It has been used in European seasonal influenza vaccines since the late 1990s. The formulation contains squalene oil mixed with citric acid buffer and stabilized with the nonionic surfactants Tween 80 and Span 85 at a concentration of 0.5% each. Originally developed by the former Chiron company, the adjuvant is now owned by Novartis Vaccines. MF59 is valued for its safety and efficacy and has been approved for human use since 1997. It has been shown to boost both humoral and cellular immunity to influenza antigens, according to a 2015 study by O’Hagan and Fox. Importantly, the adjuvant triggers a specific immune response that generates antibodies against the influenza virus, but not against squalene itself. Research suggests that MF59-enhanced influenza vaccines stimulate T-cell-dependent immunity by activating CD4+ cells and promoting isotype switching in IgG antibodies, contributing to long-lasting immunity [84]. The adjuvant also recruits various immune cells at the injection site, including macrophages, microphages, dendritic cells, and B cells. Studies have also shown increased induction of CD4 T+ follicle cells and enhanced B cell response in germinal centers of mice [85]. Comparative studies, such as that conducted by Singh et al. in 2006, suggest that MF59 has greater adjuvant potential than conventional aluminum-based adjuvants when tested with various antigens such as tetanus toxoid, hepatitis B and C, and group B and C *meningococcal* bacteria, at least in mouse models [86]. The properties of MF59 are described in Table 3.

#### 2.1.3. Immune-Stimulating Complexes (ISCOMs)

ISCOM spherical particles comprise a system of antigen delivery that is composed of cholesterol, phospholipids, saponins, and specific antigens. Immune-stimulating complexes (ISCOMs) were first reported by Morein et al. (1984) [91].


*a.* 
*Quil A*



Quil A is an aqueous extract of fractions of saponins from *Quillaja Saponaria* Molina. The significance of Quil A as a component of immune-stimulating complexes has been reported [92,93,94]. The utility of Quil A as an adjuvant in vaccine formulations has been questioned due to perceptible toxicity [95]. Recently, Skountzou et al. (2017) reviewed and reported on the safety of Quil A in humans [96]. The report showed that Quil A can induce cytotoxic (CD8+) T-cell responses when administered intramuscularly. However, Quil A has been screened as an adjuvant for skin vaccination to protect against influenza in a mouse model. The study showed that Quil A administration improved IgG and Hemagglutination-inhibition Antibody (HAI) titers. Furthermore, adjuvant Quil A administered without antigen in the dermal layer caused neutrophil infiltration, and at the injection site, the adjuvant induced cytokine networks such as IL-2, IL-6, and IL-4 production sites.


*b.* 
*QS-21*



QS-21 is an extract of the 21 active fractions of *Quillaja Saponaria*, which comprises chemically acylated 3, 28-bisdesmodic triterpene glycosides or saponins [97]. In 1991, Kensil et al. fractionized Quil A further and obtained ten fractions through the RP-HPLC method. The fractions QS-7, QS-17, QS-18, and QS-21 were found to be considerably potent as vaccine adjuvants. Among the isolated fractions, QS-18 was a major fraction but exhibited severe toxicity [98]. QS-7 and QS-21 were reported to have less toxicity [97,99]. An earlier report showed that both QS-21 and QS-7 are chemically 3,28-O-bisglycoside quillaic acid. However, QS-7 differs from QS-21, with a higher degree of glycosylation in a shorter fatty acyl unit [97]. Earlier studies have suggested that adjuvant QS-21 induces a high antigen-specific cell-mediated immune response enriched with the production of both IgG1 and IgG2a when compared to conventional aluminum hydroxide, which mainly promotes IgG1 production [94,100,101]. QS-21 has been used as an effective adjuvant in a recombinant retroviral subunit vaccine against feline leukemia virus (FeLV) because of its high potency [102]. However, the toxicity of QS-21 remains a great limitation for human use. GlaxoSmithKline demonstrated an adjuvant system (AS) using QS-21 and reported severe adverse effects [103].


*c.* 
*AS01*



AS01 is a liposome-based vaccine adjuvant system containing 3-O-desacyl-4’-monophosphoryl lipid A (MPL), with liposomes and saponin QS-21. AS01 adjuvant has been developed by GlaxoSmithKline for malarial vaccines, herpes zoster vaccines, etc. [104]. An earlier report suggested that adjuvant AS01 showed adaptive responses due to the synergistic activities of QS-21 and MPL. The adjuvant properties of AS01 showed good prospects for utilization in new vaccines targeted to populations with challenging immune status [104]. Recently, AS01 has been used as an adjuvant in a malarial vaccine (Mosquirix), which is composed of RTS, S. RTS comprises the *P. falciparum* circumsporozoite protein fused with hepatitis B surface antigen (S) [105]. AS01 has also been used in vaccine development against herpes zoster (HZ/su) [106], HIV [107], and tuberculosis [108]. An injection of AS01 resulted in the enhancement of adaptive immunity and transient innate immunity through the activation of MHC II for CD4+ cell priming, dendritic cells, neutrophils, and monocytes [104]. The activation of innate immune cells induces the release of interferon gamma (IFNγ), a macrophage-activating factor. The released IFNγ drains the lymph at the injection site. This effect results from the synergistic combination of MPL and QS-21, which is controlled by macrophages, IL-12, and IL-18. The IFNγ plays a vital role in the innate immune response against various infections and promotes Th_1_ differentiation. NK cells and cytotoxic (CD8+) T cells are the main sources of innate IFNγ.


*d.* 
*AS02*



The adjuvant AS02 is composed of MPL and QS-21 in an oil-in-water emulsion. The adjuvant was developed by GlaxoSmithKline for HIV vaccines, tuberculosis vaccines, therapeutic melanoma vaccines, and malaria vaccines [108]. Studies suggest that the AS02-adjuvanted vaccine induces strong humoral and cell-mediated responses [104]. Recently, AS02 adjuvant has been screened by using it with inactivated SARS-CoV-2 vaccine preparation [109].


*e.* 
*AS03*



GlaxoSmithKline developed the AS03 adjuvant for a pandemic flu vaccine, which is composed of α-tocopherol, squalene, and polysorbate 80 in an oil-in-water emulsion [109]. An early report suggested that the pandemic H1N1/2009 vaccine composed of AS03 showed increased immunogenicity in multiple populations when compared with non-adjuvanted H1N1 vaccines [106]. Morel et al. (2011) reported that the AS03-adjuvanted vaccine modulated the innate immune response through cytokine induction as well by enhancing the activity of monocytes that leads to antigen-specific adaptive immune responses [110,111].


*f.* 
*AS04*



The AS04 adjuvant system comprises MPL and alum. AS04 quickly induces robust and longer-lasting antibodies [112]. GlaxoSmithKline developed the AS04 adjuvant and incorporated it into hepatitis B vaccines and human papillomavirus vaccines [113]. AS04 is a toll-like receptor (TLR)-based adjuvant licensed for human vaccines. AS04 activates TLR4 and induces cytokine production through the NF-κB pathway and leads to the activation of innate immune cells. Ag-loaded dendritic cells and monocytes then lead to innate and humoral-mediated adaptive immunity [114]. GlaxoSmithKline developed AS04-adjuvanted vaccine HP16/18. In 2009, the US FDA approved AS04-adjuvanted HPV-16 and HPV-18 vaccines for human use [115]. Recently, studies have suggested that the AS04-adjuvanted HPV vaccine induces the development of neutralizing antibodies at the cervical mucosa in women aged 15–55 years [116]. Furthermore, the AS04-adjuvanted HPV vaccine induces high levels of memory B cells. Earlier reports suggested that the AS04-adjuvanted HBV vaccine demonstrated a safety profile in dialysis patients and stimulated an earlier immune response and enhanced antibody titer [117,118,119]. The advantage of AS04 is inducing a prolonged cytokine response due to the presence of aluminum salts and their action on innate immune cells through TLR4. However, TLR4 in humans does not express on lymphocytes [114].

### 2.2. Modern Adjuvant Platforms

Modern adjuvant platforms have undergone transformative advancements in immunology, opening new horizons for vaccine formulation and therapeutic applications. Following the conventional adjuvants in vaccine formulations, today’s strategies include a variety of sophisticated technologies. These include nanoparticles, often composed of lipids, proteins, or polysaccharides, tailored for precise and optimized antigen delivery that enhances specific immune responses. Virus-like particles (VLPs) mimic viral configurations without causing disease. VLPs activate the immune system by simulating the architecture of real viruses without their infectious genetic material. These breakthroughs not only improve immune responses, but also enable more precise and nuanced immunological intervention, setting the stage for the next generation of effective and tailored vaccine adjuvants to improve the efficacy of prophylactic and therapeutic vaccines. Modern vaccine adjuvants are classified as follows:

#### 2.2.1. Bacterial Derivatives

Bacterial derivatives such as metabolic products and cellular subunits have been reported as vaccine adjuvants [4]. Bacterial toxins such as cholera toxin (CT) and heat-labile enterotoxigenic *Escherichia coli* (ETEC) have been reported as mucosal adjuvants that can enhance the immune response in the mucosal layer [120,121,122]. An earlier report showed that both CT and ETEC were screened as adjuvants for nasal inactivated influenza vaccine. Bacterial cell wall components like peptidoglycan or lipopolysaccharides (LPSs) can enhance the immune response of antigens by facilitating the activation of TLRs that activate the host immune system [123,124]. Gram-negative bacteria have LPSs on the outer membrane, which are strong boosters of the innate immune system, and hence, they have significant application potential as a bacterial vaccine adjuvant. The lipid A moiety of LPSs can bind to pattern-recognition receptors such as TLR4, which leads to innate immune signaling [125]. Lipid A potentially stimulates TLR-4, which leads to intracellular signaling to activate the caspase protein system [126,127]. An earlier report suggested that the 3-O-desacyl monophosphoryl lipid A (MPL-A) series was synthesized from the 1-O-phosphono and (R)-3-hydroxytetradecanoyl moieties of *Salmonella minnesota* R595 lipid A. The MPL-A obtained from *Salmonella minnesota* was the first TLR ligand to be approved for humans as an adjuvant. MPL-A has been shown to be an innate immune cell attractant and induce pro-inflammatory cytokines. Studies suggest that MPL-A induces dendritic cell maturation, clonal expansion of CD4+ T-cells, and Th_1_ (CD3+) responses without inflammatory mediation [128]. An earlier report demonstrated the adjuvant effect of two lipophilic derivatives of muramyl dipeptide (MDP), B30-MDP and MDP-Lys(L18) B1, in a vaccine against hantavirus. That study reported that B30-MDP and MDP-Lys(L18) are useful immunoadjuvants since they induce humoral and cellular responses [129]. Another study also reported that the use of MPL-A as an adjuvant in the tetanus toxoid vaccine significantly increased specific antibodies in mice [130]. GlaxoSmithKline developed a vaccine adjuvant using MPL-A that was obtained from the cell wall LPS of Gram-negative bacteria in AS04. An earlier report suggested that the non-living cell envelopes of Gram-negative bacteria, termed bacterial ghosts (BGs), are empty cells and exhibit an immunostimulatory effect that leads to innate and adaptive immunity [131]. BGs can be produced via the controlled expression of lysis gene E of the bacteriophage phiX174 [132].

Hutter et al. (1990) reported that BGs maintain cellular morphology similar to native bacteria through electron microscopy analyses [133]. Bacteriomes are cell-derived vehicles that have been reported as promising tools in immune therapy [134]. The outer membrane vesicles (OMVs) are spherical nano-vesicles of Gram-negative bacteria that contain LPS, peptidoglycans, phospholipids, and proteins, which are usually released during the normal growth cycle [135,136]. Earlier reports suggested that OMVs have been demonstrated as a vaccine adjuvant since they stimulate a humoral-mediated immune response [136,137]. Reports suggest that BGs have been successfully used as adjuvants. BGs from V. cholerae were used as an adjuvant for antigens of *Chlamydia trachomatis* that induce localized genital mucosal Th_1_ immune responses [138]. BGs from *Escherichia coli* have been used as an adjuvant for recombinant S-layer proteins (SbsA)/outer membrane protein (Omp) 26 fusion proteins and induce an Omp260specific antibody response [139]. BGs from Escherichia coli have been screened as an adjuvant for hepatitis B core antigen and induce an HBcAg-149-specific antibody response [140]. BGs of *Salmonella typhimurium* for fimbrial antigens of ETEC elicited both humoral and cell-mediated immune responses [140]. Gong et al. (2020) reported that the OmpA of *E. coli* was used to develop novel vaccine EVP1 bacterial ghosts (EBGs) and CVP1 bacterial ghosts (CBGs) [141]. The immunogenicity of these vaccines showed protection against the hand-foot-and-mouth disease virus. Flagellin is an essential structural protein of the flagella of Gram-negative bacteria. An earlier study involved the fusion of flagellin protein for influenza A H1N1 vaccine. As a vaccine adjuvant, the advantages of flagellin have been reported as toxicity-free and not inducing allergy; hence, no IgE induction was required and minimum doses were needed for induction [142,143].

In contrast, Treanor et al. (2010) reported that flagellin showed systemic adverse effects but the induction was individualized [144]. Rajesh Mani et al. (2018) demonstrated that poly-α-L-glutamine (PLG), a cell-wall component of *Mycobacterium tuberculosis*, was an adjuvanted vaccine that displayed a robust immune response and increased cytokines, such as Th_1_-specific IFN γ, TNF-α, IL-2, Th_2_ specific IL-6, IL-10, and Th_17_-specific IL-17A cytokines [145]. An earlier report suggests that protein antigens Rv0447c, Rv2957, and Rv2958c derived from Mycobacterium tuberculosis can efficiently boost the BCG vaccine either in the presence or absence of glucopyranosyl lipid adjuvant [146]. Lactic acid bacteria (LAB) express proteins associated with PAMPs that are attracted by innate immune cells such as antigen-presenting cells (APCs), inducing intercellular communicators such as cytokine molecules [147]. LAB has been demonstrated as a mucosal vaccine adjuvant and an adjuvant for an oral vaccine against numerous viral and bacterial pathogens as well as bacterial toxins [148]. Recently, Back et al. (2019) reported that bio-components of *Mycobacterium arabinogalactan* and synthetic cord factor are components of the adjuvants. The study also reported the cell-wall skeleton of *Mycobacterium bovis* bacillus Calmette–Guérin as an effective immune stimulator [149]. Cytosine phosphor guanosine (CpG) dinucleotide motifs have been reported as a useful adjuvant for modulating immune responses. The presence of unmethylated CG dinucleotides in bacterial DNA at high frequency yields PAMPs, which attract innate immune cells [150]. CpG motifs trigger the innate immune cells that express toll-like receptor 9 to induce an innate immune response characterized by the stimulation of CD3+ (Th_1_) cells and pro-inflammatory cytokines [151]. In 2017, hepatitis B vaccine adjuvanted with CpG1018 was approved by the US—the first CpG-adjuvant vaccine approved globally [152]. CpG has been used in CoronaVac, a chemically inactivated whole SARS-CoV-2 virus particle [153,154]. Kuo et al. (2020) reported that SARS-CoV-2 spike antigen (S-2P) with CpG 1018 combined with aluminum hydroxide (alum) was found to be the most potent immunogen as a subunit vaccine against COVID-19 [155].

#### 2.2.2. Virus-like Particles

Virus-like particles (VLPs) are inactive viruses with empty capsids that lack nuclear material but preserve the virus’s structure. The desired antigens can be entrapped to VLPs using genetic engineering techniques. Several VLP-based vaccines have been licensed for the prevention of infectious illnesses such as hepatitis B virus (HBV), hepatitis E virus (HEV), human papilloma virus (HPV) influenza, and hepatitis A virus due to advancements in VLP manufacturing, purification, and adjuvant optimization [156,157,158,159]. Human papilloma virus vaccine has been created using the VLP technique and licensed by the US Food and Drug Administration for clinical usage [23]. Recombinant hepatitis B surface antigen (HBsAg) has been generated as VLPs in *Saccharomyces cerevisiae* [160]. Müller et al. (2020) reported that VLP-based immunization strategies may represent a powerful approach for generating polyclonal sera containing cross-reactive neutralizing antibodies against *Lassa mammarenavirus* (LASV) [161]. LASV GP-induced VLPs were shown to be immunoreactive in an animal model and capable of producing robust broad neutralizing antibody (nAb) when combined with a squalene-based adjuvant formulation. The predominant target of neutralizing antibodies against LASV is the envelope surface glycoprotein complex (GP), which is also the primary viral antigen employed in the construction of an LASV vaccine. VLP-based vaccination techniques could be an effective way to generate cross-reactive neutralizing polyclonal antibodies against LASV. VLPs are known as subunit vaccines and are divided into enveloped and non-enveloped subtypes. They are used as vaccines to prevent infectious diseases and cancers by activating the humoral and cellular immune systems [162].

VLP viral structures and morphologies are advantageous for immunostimulatory function because they are more efficiently recognized by APCs such as dendritic cells, resulting in a powerful immune response by trafficking from the injection site to the lymph nodes. Virus structural antigens stimulate B-cell activation, which results in a higher humoral immune response and cellular-mediated immunity [163,164]. Adjuvant carriers such as aluminum salts may not be required for VLP-based vaccines. Indeed, adjuvant formulations for VLP-based vaccinations are currently moving away from aluminum salts in favor of other methods. Virosomes are vesicular particles derived from viral envelopes that are suitable for delivering antigens and DNA, and drugs are encapsulated in the empty virosome lumen. Virosomes have been utilized to deliver vaccines because they are noninfectious, adjuvant-free, biodegradable, and do not replicate in the host. Virosomes can deliver antigens to APCs through MHC class II and MHC class I, capable of and eliciting a balanced Th_1_ and Th_2_ immune response, and can induce cytokines such as TNF-α, IFN-γ, and IL-2 [165,166]. Numerous forms of virosomes have been used to administer vaccines, including HIV, influenza, and non-influenza viral virosomes, hepatitis virus virosomes, and Sendai virus virosomes [167]. Virosomes can merge with the plasma membrane at a neutral pH or with the endosomal membrane at an acidic pH following receptor-mediated endocytosis [168]. After supplementing virosomes with cationic lipids such as dioleoyloxypropyl-trimethylammonium methyl sulfate (DOTAP) and dioleoyl dimethylammonium sulfate (DMAS), the unique targeting features of spike proteins were used in the creation of DNA virosomes [168]. Virosomes are virus-like encapsulated particles that allow for fine control over particle composition and adaptability to different antigens. Virosomes have carrier and immune-stimulatory properties. Virosomal technology is FDA-approved and has a high safety profile. Virosomes are non-toxic, biocompatible, and biologically degradable. However, virosomes are constrained by manufacturing issues, instability, and fast disintegration [169,170]. Influenza virosomes are utilized in commercial vaccinations. The technology has been used as a carrier and adjuvant for subunit vaccinations, especially synthetic peptides [171]. Recently, a study in respect of a virosome vaccine containing the beta spike protein and comparing its immunogenicity in mice to a virosome vaccine containing the original Wuhan spike found that the Wuhan spike vaccine was more effective in producing an immune response. According to the findings of the study, beta spike immunization results in less effective neutralization [172]

#### 2.2.3. Bacteriophages as Vectors

With the introduction of phage vaccines, it was demonstrated in 1985 that it was possible to produce bacteriophages that exhibit foreign proteins fused to their regular coat proteins [173]. The power of phage display technology was significantly increased by utilizing affinity selection to separate out specific peptide-displaying phages from random peptide collections. The use of this technology to create vaccinations and diagnostics has taken off as an industry. It is also feasible to use epitopes that have been selected based on biological research [174,175,176]. Additionally, the utilization of epitopes that are selected based on biological research is a viable option. For example, the use of a phage display system in a vaccine was able to provide complete immunological protection to mice that were nasally challenged with a live respiratory syncytial virus [177,178]. Bacteriophage-based vaccinations are thought to be a strong replacement option for conventional vaccines, due to the drawbacks of conventional vaccines [179]. This strategy makes use of characteristics that are intrinsic to bacteriophages to improve the stability and immunogenicity of the antigens that are presented. This tactic makes use of the potential of phage particles to induce both cellular and humoral immune responses, which is an advantage to the overall strategy.

#### 2.2.4. Liposomes

Liposomes are vesicles of variable sizes that are created in vitro and consist of a spherical lipid bilayer and an aqueous inner compartment [180]. Liposomes, a contemporary vesicular antigen delivery method that overcomes the drawbacks of old or conventional antigen delivery methods, can prolong vaccination release. Liposomes are often made up of different types of amphiphilic phospholipids, such as phosphatidylcholine, phosphatidylserine, and sphingomyelin, which can be coupled with other lipids, such as cholesterol, for membrane stabilization, and negatively or positively charged lipids to regulate liposome structure and surface properties. As a vehicle for the delivery of vaccines, liposomes offer several benefits, the most important of which are the capacity to prevent the degradation of antigens, transport single or multiple hydrophilic and lipophilic antigens, regulate the release of antigens, improve antigen-specific immune responses, and enhance cellular uptake (Table 4).

Liposomes can have their physicochemical qualities altered to improve their capacity to target antigen-presenting cells. These alterations can be made to the size, charge, and membrane fluidity of the liposomes. To create effective vaccines based on liposomes, it is essential to have a thorough understanding of the ways in which the various physicochemical features of liposomes influence the overall immune response [177]. Liposome characteristics have a crucial impact in local tissue distribution, retention, trafficking, uptake, and processing by APCs, even though most vaccinations are delivered through intramuscular or subcutaneous injection. Previous research demonstrated unequivocal size-dependent effects at the injection site; however, there was no clarity on charge or lipid composition-dependent effects [190]. The cationic liposome formulation dimethyl dioctadecyl ammonium (DDA) with trehalose dibehenate (TDB) (DDA/TDB, CAF01) used in earlier experiments did not reveal any changes in liposome draining or antigen release at the injection site. However, there were notable disparities in the movement to the regional lymph nodes [191]. Liposomal vaccines against influenza, hepatitis A, malaria, and Varicella zoster virus are commercially available. Several liposomal formulations are undergoing clinical trials as preventative and therapeutic vaccinations against malaria, influenza, tuberculosis, HIV, and dengue fever [192]. Flexibility and versatility define liposomes that can encapsulate hydrophilic and lipophilic substances. Due to their vesicular shape, liposomes can be customized to provide the desired immune response and adjuvant properties. Many liposome-based vaccinations were successful during the COVID-19 pandemic [193]. The development of improved and subunit-defined nano-vaccines shows promise in several immune responses mediated by liposomes in vivo [194]. Liposomes have been shown to be endocytosed and to go through the processing of a highly well-characterized endocytic pathway, which results in the delivery of several antigens to lysosomes [195]. Liposomes are important delivery systems because they can modulate the generation of CD4+- and CD8+-mediated immune responses (Figure 2), as well as the generation of Th_1_, Th_2_, or Th_1_/Th_2_ phenotypes, and they may also modulate a Th_1_/Th_2_ switch. This makes liposomes one of the most versatile delivery systems available [186].

#### 2.2.5. Nanosomes

Nanosomes are lipid-based nanoparticles that can be used as adjuvants in vaccine formulations. Nanosomes are composed of lipid bilayers, similar to cell membranes, and can encapsulate or carry various vaccine antigens. They are similar in structure to liposomes but are composed of phosphatidylcholine and phosphatidylethanolamine at a 3:1 ratio and have a smaller size range (20–50 nm). Like other lipid-based adjuvants, nanosomes can enhance antigen delivery and promote uptake by antigen-presenting cells, leading to enhanced immune responses. Nanosomes have been used in several vaccine formulations, including vaccines against influenza, hepatitis B, and human papillomavirus. Studies have shown that nanosome-based vaccine formulations can enhance both antibody and cellular immune responses and provide protective immunity against viral infections [196].

#### 2.2.6. Nanoparticles

Biodegradable particulate vaccine delivery systems use biodegradable polymeric articles as carriers for antigens, adjuvants, or both. These particles are designed to enhance the delivery, stability, and immunogenicity of the vaccine. Biodegradable polymeric microparticles can indeed be used as vaccine adjuvants. Biodegradable polymers are used as sutures and drug carriers due to their non-toxicity and tunable biodegradability. Currently, biodegradable polymers represent a class of ubiquitous materials that are used for a variety of purposes due to the increasing interest of the pharmaceutical industry in the production of drug delivery systems. The polymers selected as excipients for parenterally administered particles should meet several requirements, including being biodegradable, safe (tissue compatible, no secondary reaction), drug resistant and permeable, stable in vitro, easy to process, having formulations that are responsible on their own, and ideally cost effective. Over the past two decades, the idea of drug and vaccine delivery systems has evolved significantly. In the area of controlled release technology, biodegradable polymers offer the ability to sustain the effects of drugs and vaccines over extended periods of time. Controlled release of vaccines from a biodegradable polymer matrix is one of the most promising delivery techniques currently under investigation [197]. APCs have efficiently taken up and processed antigens, enhancing cross-presentation via the major histocompatibility complex of class I and class II (MHC I and II), leading to the induction of CD4+ cell and CD8+ T cell responses. The advantages of polymeric biomaterials are that they can increase the stability of vaccine antigens through encapsulation or adsorption by inhibiting their degradation [109], as well as through the combination of different adjuvants and antigens, which ultimately increases the immunogenicity of less potent vaccine antigens. Their adaptability in surface functionalization enables active targeting of APCs through the activation of endosomal TLRs or surface PRRs, both of which lead to robust immune responses [197]. Polymeric microspheres have shown remarkable potential as a next-generation adjuvant to replace or supplement existing aluminum salts and enhance vaccination [198].

Biodegradable polymeric nanoparticle systems have been explored as a promising platform for vaccine delivery [199]. Biodegradable polymeric nanoparticles have shown great potential as vaccine adjuvants in recent years. These nanoparticles can enhance the immune response to vaccines by improving antigen delivery, promoting antigen uptake by immune cells, and modulating the immune system’s activation [199]. Polymeric nanoparticles can encapsulate antigens within their structure, protecting them from degradation and enhancing their stability. This encapsulation allows for controlled release of the antigens, leading to sustained antigen exposure to the immune system. The nanoparticles can target antigen-presenting cells, such as dendritic cells, promoting efficient antigen uptake and initiating a robust immune response [200]. Biodegradable nanoparticles can be engineered to possess specific properties that allow for the modulation of the immune response. By manipulating various characteristics of the nanoparticles, researchers can tailor their interactions with the immune system and fine-tune the resulting immune response. The size and surface charge of nanoparticles can influence their interaction with immune cells and their trafficking within the body. For example, smaller nanoparticles can more effectively penetrate tissues and be taken up by immune cells. The surface charge of the nanoparticles can also affect their cellular uptake and interaction with proteins and receptors on immune cells. The nanoparticles can deliver antigens to APCs such as dendritic cells (DCs). The APCs take up the antigens and are processed into smaller fragments, which are then presented on the cell surface using major histocompatibility complex (MHC) molecules. This presentation triggers the activation of T cells and initiates an immune response [201]. The nanoparticles themselves or the adjuvants incorporated into them can have immunostimulatory properties. They can activate immune cells, such as macrophages and dendritic cells, leading to the secretion of cytokines and chemokines and the recruitment of other immune cells to the site of nanoparticle administration [202,203].

A nanoparticle-adjuvanted vaccine is designed to be recognized by various components of the innate immune system, such as dendritic cells, macrophages, and natural killer cells. This recognition occurs through pattern-recognition receptors (PRRs) present in these cells, which recognize pathogen-associated molecular patterns (PAMPs) or danger-associated molecular patterns (DAMPs) displayed on the nanoparticles [202,204,205,206]. Interaction between vaccine nanoparticles and innate immune cells triggers the release of cytokines, chemokines, and other inflammatory mediators [207]. This activation leads to the recruitment of immune cells to the administration site. Nanoparticle-adjuvanted vaccines can enhance antigen uptake and processing by APCs. They can be designed to release antigens in a controlled manner, ensuring prolonged exposure and better antigen presentation to initiate an immune response (Figure 3). Adjuvants stimulate APCs, promote the secretion of cytokines, enhance antigen presentation, and improve the overall immune response. The antigens displayed on MHC molecules on APCs activate antigen-specific T cells, including helper T cells (CD4+) and cytotoxic T cells (CD8+). Vaccine nanoparticles can stimulate B cells to produce antibodies. Antigen presentation by APCs triggers B cell activation, resulting in antibody production. The sustained release of antigens from nanoparticles can provide prolonged stimulation, leading to robust and long-lasting antibodies [207,208,209].

#### 2.2.7. Nanovesicles

Vaccine delivery systems based on fatty acid nanovesicles offer several advantages, such as biocompatibility and biodegradability, making them safe for vaccine delivery [210]. Fatty acids are present in several human tissues and fluids. They are stored in the cytoplasm of cells and serve as an important source of energy. Fatty acids play an essential role in biological processes and are thought to be found throughout the body. More specifically, they are stored in the cytoplasm of cells and serve as an important source of energy. Nanovesicles of fatty acids can protect encapsulated vaccines from degradation, provide controlled release kinetics, and allow targeted delivery to specific cells or tissues. By modifying the surface of the nanovesicles, such as by attaching ligands, they can be targeted to specific receptors on immune cells, enhancing vaccine uptake and immune activation (Table 5). Omega-3 fatty acids are a component of the skin, and a vesicular omega-3 fatty acid formulation will have a high degree of membrane binding and be particularly biocompatible with the cellular system. The recognition that many commonly used live vaccines elicit immune responses through activation of pattern recognition receptors was made possible by significant conceptual advances in innate immunity and the explanation of the key innate receptor, the pattern recognition receptor. As a result, the cytokine network is targeted by identifying cells of the innate immune system to activate CD4+ and CD8+ cells (Figure 4). Studies have demonstrated the potential of docosahexaenoic acid nanovesicles for the delivery of recombinant HBsAg. Encapsulation of HBsAg in docosahexaenoic acid nanovesicles has been shown to increase its stability and immunogenicity, resulting in better antibody responses compared to conventional vaccine formulations [207,210].

## 3. Future Trends

Significant progress has been made in the field of vaccine adjuvants, as adjuvants play a critical role in enhancing the efficacy, longevity, and specificity of vaccine-induced immune responses. Toll-like receptors (TLRs) are a class of proteins that play a key role in the innate immune system and serve as primary sensors for pathogens. By recognizing pathogen-associated molecular patterns (PAMPs), TLRs activate immune responses and set the stage for adaptive immunity. The innovative adjuvant, a polymeric TLR 7 agonist in the form of a nanoparticle (TLR7-NP), is characterized by targeting lymph nodes and enhancing immune cell activation. In combination with alum-bound antigens, it stimulates the production of antibodies targeting both major and less dominant epitopes and enhances the response of CD8+ T cells in mice. This boosted influenza vaccine protects mice against multiple influenza strains. Integration with a SARS-CoV-2 vaccine also broadens the defense against novel viral mutations. Because of their ability to link innate and adaptive immunity, TLR agonists have great potential as adjuvants in vaccines to combat serious diseases such as cancer, AIDS, and malaria. The potential of combining TLR7/8 agonists with both current and novel antigens will be crucial adjuvant development for vaccines against cancer, infectious diseases, and allergic and autoimmune disorders. R837, also known as imiquimod, is a synthetic compound recognized for its immune-modulating properties. Primarily known as a treatment for certain skin conditions like actinic keratosis, superficial basal cell carcinoma, and genital warts due to its ability to stimulate the immune response, R837 also acts as a TLR7 agonist [215,216]. Because of its mode of action, it has been explored as a potential vaccine adjuvant. Dendritic Cell-Based Tumor Vaccine Adjuvant Immunotherapy for glioblastoma multiforme is under clinical trials [217]. TLR 4 stimulant, glucopyranosyl lipid adjuvant in a stable emulsion, (GLA-SE) promotes robust TH_1_ responses and balanced IgG_1_/IgG_2_ responses to protein vaccine antigens. This enhanced immune response provides adequate protection against numerous diseases, including tuberculosis and leishmaniasis. GLA-SE is an adjuvant currently undergoing clinical evaluations. Research indicates that it significantly boosts Th_1_ immune responses [218]. Currently, the primary emphasis of regulatory action is to ensure the safety and efficacy of products, as adverse reactions can have a significant impact on public health. Challenges include several aspects, namely establishing factual immune response enhancement, ensuring consistent production quality, the lack of standardized evaluation measures, and resolving the complex issues associated with combination vaccines where adjuvants may interact with multiple antigens.

## 4. Conclusions

Vaccine adjuvants play a critical role in enhancing the immune response to vaccines and improving their efficacy. The evolution of vaccine adjuvant technology has been a profound journey, encapsulating the strides of scientific discovery from the early days of alum to the modern era’s intricate nanoformulations. Alum, with its long-standing history, laid the foundational understanding of adjuvanticity, leveraging its ability to enhance certain humoral immune responses. However, as the complexities of immunity unraveled, the limitations of alum became more evident, especially in its capacity to stimulate a comprehensive cellular immune response. The field of vaccine adjuvants has evolved from traditional alum-based formulations to more advanced nanoparticle-based formulations. These advances have enabled the development of vaccines with improved efficacy and safety profiles. In addition, nanoparticle-based adjuvants have shown promise in enhancing immune responses by improving antigen delivery, stability, and immune cell activation. A recent study has shown that docosahexaenoic acid (DHA) nanovesicles are a promising adjuvant for recombinant HBsAg, as they stimulate innate and adaptive immunity. As research and development continues, we expect to see further innovations in the field of vaccine adjuvants to optimize immune responses and protect against various diseases. The future of vaccine adjuvants will be driven by personalized approaches, computational modeling, advanced delivery systems, combination adjuvants, therapeutic applications, and robust safety assessment.

## Figures and Tables

**Figure 1 vaccines-11-01704-f001:**
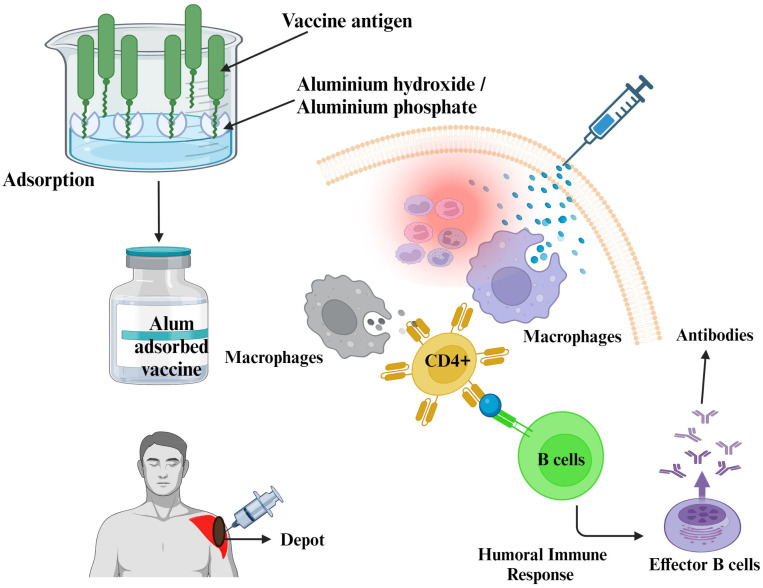
The mechanistic approach of aluminum salts as vaccine adjuvants. This figure was Created with BioRender.com, Bio Render, Canada.

**Figure 2 vaccines-11-01704-f002:**
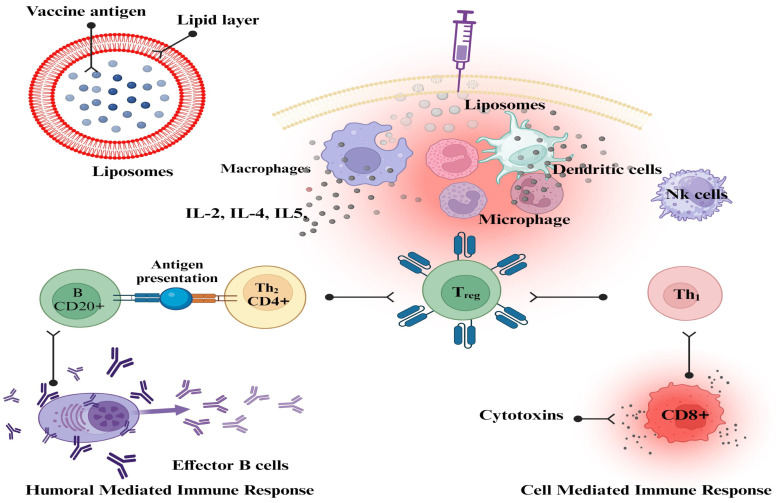
The mechanistic approach of liposomes as vaccine adjuvants. This figure was Created with BioRender.com, Bio Render, Canada.

**Figure 3 vaccines-11-01704-f003:**
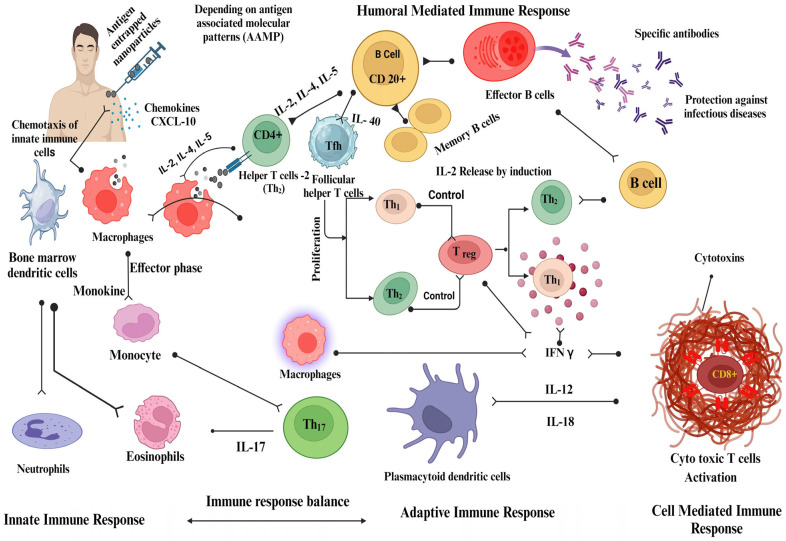
The mechanistic approach of nanoparticles as vaccine adjuvants. This figure was created with BioRender.com, Bio Render, Canada.

**Figure 4 vaccines-11-01704-f004:**
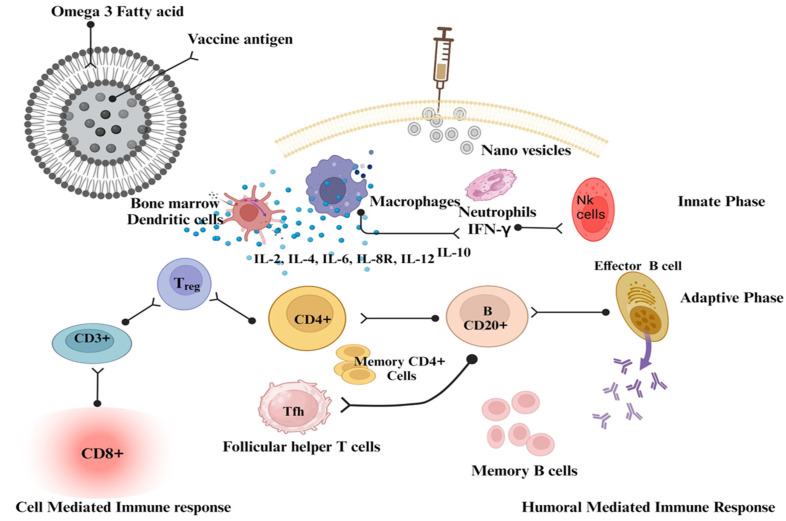
The mechanistic approach of nanovesicles as vaccine adjuvants. This figure was created with BioRender.com, Bio Render, Canada.

**Table 2 vaccines-11-01704-t002:** Types of calcium adjuvants for suitable vaccines.

Calcium Adjuvant	Characteristics	Examples of Vaccines	References
Calcium phosphate	White, crystalline powder	Anthrax, Diphtheria–Tetanus–Pertussis, Haemophilus influenzae type b	[31,53]
Calcium carbonate	White, chalky powder	Human Papillomavirus	[53]
Calcium chloride	White, crystalline powder	Rabies	[31,53]
Calcium gluconate	White, crystalline powder	Influenza	[54,55]

**Table 3 vaccines-11-01704-t003:** Functional properties of MF59 as a vaccine adjuvant.

Properties	References
Upregulation of cytokines, MHC class II, and costimulatory molecules, and promotion of DC migration to the T cell area of the lymph node. Stimulation of monocyte attraction, attraction of CD11b+ cells, and improved antigen uptake by dendritic cells.	[87]
2. The adjuvant properties of MF59 are reported to be fully unrelated to the NOD-like receptor (NLR) family, which are sensors in innate immune cells. The NALP3 inflammasome is a complex formed by a specific pattern-recognition receptor in the cytoplasm. MF59 depends on myeloid differentiation factor 88 (MyD88) for enhancing immune responses, irrespective of TLR pathways. MyD88 plays a pivotal role in the signaling of toll-like receptor/interleukin-1 receptor, leading to the activation of nuclear factor-kappa B.	[88]
3. The MF59 adjuvant activates both humoral and cell-mediated immunity through Th_1_ and Th_2_ induction. It outperforms alum-based adjuvants when used in influenza vaccines. It exhibits an enhanced immune response as evidenced by the increased presence of hemagglutination inhibitory (HAI) antibodies and memory T and B cells, particularly against antigenically distinct influenza strains. This leads to more effective vaccines for both pandemic and regular seasonal influenza.	[89]
4. The most frequently observed local reaction is mild to moderate pain at the injection site. Erythema and induration also noted.	[90]

**Table 4 vaccines-11-01704-t004:** Functional properties of liposomes as vaccine adjuvants.

Properties	References
Liposomes can protect antigens from degradation and prolong their longevity. By encapsulating the antigen molecules, liposomes protect them from enzymatic degradation, increasing their stability in the body.	[178]
2. Liposomes can enhance the body’s immune response to the introduced antigen. By transporting antigens to antigen-presenting cells such as dendritic cells, liposomes promote better antigen presentation to T cells.	[179]
3. They are biologically compatible, harmless, and capable of breaking down naturally. Liposomes can simultaneously encapsulate antigens and molecular adjuvants, ensuring their delivery to the same immune cells, resulting in a stronger immune reaction. Liposomes are particularly adept at targeting dendritic cells, which are crucial in triggering immune responses.	[180]
4. Specific kinds of liposomes can directly stimulate the NLRP3 inflammasome. Cationic liposomes that activate the NLRP3 inflammasome may enhance antigen presentation. Specific cationic liposomes can trigger activation of the NLRP3 inflammasome in macrophages, suggesting their possible use as immunostimulants.	[181,182,183,184]
5. Liposomes, when used in vaccines or immunotherapies, can stimulate both innate and adaptive immune responses. By carrying pathogen-associated molecular patterns (PAMPs) or specific immunomodulatory molecules, liposomes can activate key players of the innate immune system such as dendritic cells, macrophages, and NK cells. In addition, by efficiently delivering and presenting antigens, liposomes can enhance the adaptive immune response by activating specific T and B lymphocytes. This dual action provides comprehensive immune defense and makes liposomes valuable tools for vaccine development.	[185,186,187,188,189]

**Table 5 vaccines-11-01704-t005:** Functional properties of fatty acid nanovesicles as vaccine adjuvants.

Properties	References
The application of nanovesicles derived from fatty acids as vaccine adjuvants represents a promising area of research in the field of immunology, mainly owing to their intrinsic biocompatibility. Fatty acids, inherent constituents of the human body, are readily acknowledged and digested without inducing substantial adverse effects. Consequently, they represent highly suitable candidates for the development of vaccination formulations. The capacity to integrate with cellular membranes, enhance antigen absorption, and permit extended antigen presentation highlights the potential effectiveness of these entities. In addition, the biodegradable properties of these substances prevent their accumulation within the human body, hence preserving a crucial safety profile necessary for the effective utilization of vaccines.	[207,210,211]
2. Fatty acids have been shown to play an important role in regulating inflammasomes, a complex intracellular machinery responsible for triggering inflammatory responses. Different types of fatty acids can have different effects on inflammasome activation. For example, saturated fatty acids such as palmitic acid are known to promote inflammasome activation, leading to the production of pro-inflammatory cytokines such as IL-1β. This activation may contribute to diseases such as insulin resistance and atherosclerosis.	[207,212]
3. Omega-3 fatty acids, especially eicosapentaenoic acid (EPA) and docosahexaenoic acid (DHA), are known for their anti-inflammatory properties. One of the methods by which the observed effect is achieved is through the regulation of NLRP3 inflammasome activation. The NLRP3 inflammasome is a complex consisting of many proteins involved in the synthesis of pro-inflammatory cytokines such as IL-1β and IL-18. These cytokines are known to play important roles in several inflammatory diseases.	[213]
4. The incorporation of omega-3 fatty acids in vaccine formulations may provide a dual benefit: it increases vaccine efficacy by promoting a well-regulated immune response and potentially mitigates excessive inflammation caused by specific adjuvants. Thorough studies are needed to determine the appropriate dosage, efficacy, and safety of this technique. However, the potential synergistic effects of omega-3 fatty acids and vaccine adjuvants present an interesting avenue to advance vaccine development in the future.	[214]

## Data Availability

Data sharing is not applicable to this article.

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
