# Peer review of "Advancements in Vaccine Adjuvants: The Journey from Alum to Nano Formulations"

_vaccines, 2023, doi:10.3390/vaccines11111704_

Round 1
Reviewer 1 Report
Comments and Suggestions for Authors
Review report
The manuscript titled “Advancements in Vaccine Adjuvants: Journey from Alum to Nano formulations” authored by S S Moni et.al reviews the existing evidence on various adjuvants and attempts to give an overview of the history and evolutions of adjuvants. It is appreciated that authors have done a commendable effort in gathering relevant information pertaining to the filed and have tried to concise that into an article, which is highly appropriate for Vaccines journal. My recommendation would be “Major revision” to this article. While I have no major concerns or comments on the information collected and the message the article is trying to deliver, I have a strong opinion on how poorly it is written down and presented. The following is a non-exclusive list of recommended changes- just a directive on what needs to be changed.
· In my opinion, more efforts need to be given to organization of the information and clarity of presentation. The article in the current form lacks flow of information. While all the information required is there, they are scattered and make it difficult for the reader to comprehend.
For example: Line 50 describes about the vaccine invention, line 51 about prophylactic and therapeutic vaccines while line 54 speaks about cost of vaccination. This is just an example of how the article has no gradual flow of information. Moreover, all three sentences (and more such ones throughout) mentioned need to be restructured to improve clarity.
Lines 64-78 speaks a lot about adjuvants, their use, mode of action etc while the term ‘adjuvant’ itself is introduced and described much later in line 110. This is a perfect example of lack of flow of information.
· Line 113-116: It is mentioned that adjuvant discovery is “accidental” and ‘Ramon found that horses that developed an abscess at injection site had higher specific antibody titres….” However, it is not clear how this observation led to adjuvant discovery. Please explain.
· Line 239: Reads “b. Calcium phosphate”. Which one is “a.” ?
· Similarly Line 274 : Reads “2.Oil emulsion adjuvants” . Which one is “1.”?
· It would highly improve the organization of the article if a numbering system were followed with each category getting a number (eg 2. Oil emulsion adjuvants) followed by sub-numbering the members in that category (2.1 CFA, 2.2 IFA, 2.3 Montanide etc) This pattern can be applied for all categories of adjuvants mentioned. Alternately, there could be table/ chart with these categories wee-represented- I leave it to
· ‘Adjuvants are meant to be for enhancing immune response against the vaccine-targeted antigen.’ Once that statement is made (Line 108) (in my opinion should have mentioned much earlier in the manuscript) the repetition of the same could be avoided. (Line 310 etc)
· There is a clear distinction between a vaccine strategy, vaccine delivery system and an adjuvant, which I believe the authors have missed completely. There is a whole section on Virus-like particles(VLPs). VLPs are neither delivery vehicles nor adjuvants but are in fact vaccine antigens themselves (Eg HPV VLP vaccine is a mix of antigens different types of HPVs together). VLP based vaccines have adjuvants included the formulation (eg. Amorphous aluminium hydroxyphosphate sulfate is used as adjuvant in HPV vaccine Gardasil). It is unclear why a section on VLPs is included under the ‘Types of adjuvants’. VLPs is a vaccine strategy and not an adjuvant system. This section seems to be a total misfit in the manuscript. I have the same opinion on section “Bacteriophage as vector”. This again is a vaccine strategy and not adjuvant system. Again, a misfit.
As mentioned earlier, this is not a complete list of corrections expected from the authors, but just a directive on what is missing and what needs to be improved. In my opinion, this manuscript can be improved, provided adequate efforts are made in re-organizing the article along with professional language edits.
Comments on the Quality of English Language
· This article needs to be thoroughly edited by an English language professional before it could be published in any journal. There are extensive simple grammar mistakes throughout the article (I have listed down very few of them) and sometimes totally alter the meaning of some sentences (eg: Lines 79, 205 etc). Moreover, in the present form, the article lacks a scientific tone.
Line 40: Need to add ‘who’ to make the sentence correct.( ……English physician ‘who’ contributed to ….)
Line 43: “Edward Jenner had been observed…” is wrong. The right format would be “Edward Jenner had observed….” or simply “ Edward Jenner observed…..”
Line 45: “This laid Jenner…..” is incorrect. The right word to be used there is ‘led’ and the sentence should read “This led Jenner….”
These are just a couple of examples, there are several more, but I would not list all of them here and would leave it to the authors to take more efforts themselves or invite support from professionals in language editing. Taking these efforts alone would significantly improve the quality of the article.
Author Response
comments attached

Reviewer 2 Report
Comments and Suggestions for Authors
The manuscript by Sivakumar S.M. et al., entitled “Advancements in Vaccine Adjuvants: Journey from Alum to Nano formulations” summarized the evolution of vaccine adjuvants in this review. It is a comprehensive summary of adjuvants. However, the enthusiasm of the review is dampened by poor English. A thorough editing is required. There are few minor concerns (listed below) which need to be addressed. Over all, this manuscript is recommended for minor revision before its acceptance.
Minor concerns:
1. Appropriate figures of each adjuvant category should be included for a better understanding.
2. A mechanistic figure of vaccine adjuvants, would be helpful for the readers.
3. The authors should provide one table comprising the adjuvants are being used in clinical trials currently.
Comments on the Quality of English LanguageQuality of English in the manuscript should be improved significantly. Grammatical errors are there in multiple sentences.
Author Response
response to comments

Reviewer 3 Report
Comments and Suggestions for Authors
Advancements in Vaccine Adjuvants: Journey from Alum to 2 Nano formulations
Manuscript ID: vaccines-2676992
The authors have summarised the various types of adjuvants used in vaccines in a very precise and interesting way. All the important information related to the topic is presented in a well-organised way. The authors have discussed
Major comments:
1. Please include more precise tables
2. Include figures in the manuscript
Author Response
Response to comments

Round 2
Reviewer 1 Report
Comments and Suggestions for Authors
I appreciate the efforts undertaken by the authors to respond to the comments and make appropriate changes to the manuscript after the first round of review. Authors have addressed most of my comments from the previous report and the manuscript can be accepted after a couple of minor corrections.
There is only one comment from the earlier review report that was missed out by the authors during revision and is given below.
Line 113-116: It is mentioned that adjuvant discovery is “accidental” and ‘Ramon found that horses that developed an abscess at injection site had higher specific antibody titres….” However, it is not clear how this observation led to the accidental discovery of adjuvants. Please explain.
Regarding my previous comment on including sections on VLPS - I totally agree with the authors on the “necessity to analyze VLP and bacteriophages”. However, since VLPs are primarily a vaccine strategy and not adjuvant, including them as a “Type of adjuvant” would not be appropriate. Instead, authors could improvise by including a section on “adjuvants used not only in infectious vaccines but also in DNA vaccines” for better fitting the topic. Now, with proper numbering system in the revised format it fits slightly better.
Author Response
EXPLANTION TO REVIEWER – 1, Round 2
Respected Editor,
Greetings. Thank you very much for reviewing.
Comment 1:
I appreciate the efforts undertaken by the authors to respond to the comments and make appropriate changes to the manuscript after the first round of review. Authors have addressed most of my comments from the previous report and the manuscript can be accepted after a couple of minor corrections.
Response: We greatly appreciate your comments, It is highly encouraging for us to develop further in the pathway of research.
Comment 2: There is only one comment from the earlier review report that was missed out by the authors during revision and is given below.
Response: We greatly appreciate your comments, and I would like to express our commitment to addressing each of your suggestions individually by revising and improving our manuscript, as per the recommendations of the reviewer.
Comment 3: Line 113-116: It is mentioned that adjuvant discovery is “accidental” and ‘Ramon found that horses that developed an abscess at injection site had higher specific antibody titres….” However, it is not clear how this observation led to the accidental discovery of adjuvants. Please explain.
Response: Thank you for your comments. You're absolutely right. I had read about it previously and had prewritten thoughts on it. I am sorry about it. The information has now been changed, updated, and highlighted in yellow.
Comment 4: Regarding my previous comment on including sections on VLPS - I totally agree with the authors on the “necessity to analyze VLP and bacteriophages”. However, since VLPs are primarily a vaccine strategy and not adjuvant, including them as a “Type of adjuvant” would not be appropriate. Instead, authors could improvise by including a section on “adjuvants used not only in infectious vaccines but also in DNA vaccines” for better fitting the topic. Now, with proper numbering system in the revised format it fits slightly better.
Response: Thank you for your suggestions. They're excellent, and I greatly appreciate them. Yes, as suggested we have changed as Types of adjuvants for both prophylactic and DNA vaccines.
Thank you very much